# Buckling of Corrugated Ring under Uniform External Pressure

**Igor I. Andrianov [1,\*], Igor V. Andrianov [2], Alexander A. Diskovsky [3] and Eduard V. Ryzhkov [3]**

[1] Rhein Energie AG, Parkgürtel 24, D-50823 Köln, Germany
[2] Institut für Allgemeine Mechanik, RWTH Aachen University, Templergraben 64, D-52056 Aachen, Germany; igor_andrianov@hotmail.com
[3] Department of Economic and Information Security, Dnipro State University of Foreign Affairs, Gagarina Ave 26, UA-49005 Dnipro, Ukraine; alex_diskovskiy@ukr.net (A.A.D.); revord@ukr.net (E.V.R.)
[\*] Correspondence: i.andrianov@rheinenergie.com

**Abstract:** Stability analysis of a corrugated ring subjected to uniform external pressure is under consideration. Two main approaches to solving this problem are analyzed. The equivalent bending stiffness approach is often used in engineering practice. It is based on some plausible assumptions about the behavior of a structure. Its advantage is the simplicity of the obtained relations; the disadvantage is the difficulty in estimating the area of applicability. In this paper, we developed an asymptotic homogenization method for calculating the critical pressure for a corrugated ring, which made it possible to mathematically substantiate and refine the equivalent bending stiffness approach. To evaluate the results obtained using the equivalent stiffness approach and asymptotic homogenization method, the imperfection method is used. The influence of the corrugation parameters on buckling pressure is analyzed.

**Keywords:** stability; corrugated ring; external pressure; equivalent bending stiffness approach; asymptotic homogenization method; imperfection method

## 1. Introduction

Corrugation of smooth shells is widely used to change their stiffness. In particular, for cylindrical shells under external pressure, the corrugation of the generatrix is used [1–7]; corrugation of the directrix significantly increases the stability of such shells under axial compression [8]. Therefore, the study of the stability of corrugated shells under external pressure is of practical importance.

The methods currently used to calculate corrugated plates and shells can be subdivided into three groups: computational models based on finite element analysis (FEA), the equivalent bending stiffness approach, based on some physical assumptions, and the asymptotic homogenization method.

FEA in principle makes it possible to calculate any corrugated shell [6–15], but, for the early stages of design, simple analytic solutions are very useful to engineers. Analytical solutions can be also used as benchmark examples for numerical algorithms.

Before the computational revolution, the main method for calculating corrugated shells was equivalent stiffnesses modeling [16,17]. A common approach is to replace a corrugated shell with an anisotropic one that has equivalent stiffness properties. This method has been used in the analysis of corrugated shells since the 1930s [17]. The main disadvantage of this approach is the difficulty in estimating accuracy and the area of applicability [18–26].

In other papers [27–31], the asymptotic homogenization method was used. The original equations of the corrugated shell are projected on the basic surface, equally spaced from the corrugation crests. This allows the establishment of a relation between the homogenized and real components of the



stress–strain state. In the zeroth approximation, the slowly changing components of the solution are determined. The subsequent approximations are rapidly oscillating periodic functions (correctors) and they are found in the solution to the cell problem. This approach allows the solution of a number of optimization problems and the consideration of structures with functionally graded corrugations [29–31].

The buckling problem for longitudinally corrugated cylindrical shells under external pressure was investigated by Semenyuk and Babich [9]. They analyzed simply supported cylindrical shells, of which cross sections are described in polar coordinates by Fourier series $R(\varphi) = R_0 + \sum\limits_{k=1} R_k \cos(k\varphi)$. The solution was represented as the product of the sine in the longitudinal direction on the Fourier series in the circumferential direction. Numerical results show the decrease in critical pressure for a corrugated shell in comparison with a cylindrical shell of the same length and radius $R_0$.

The nonlinear elastic deformation of a thin flexible ring under the external hydrostatic pressure is described in [15].

In our work, we study the stability of the corrugated ring subjected to external pressure. Obtained results can be used for long cylindrical shells, when the number of waves in the circumferential direction is two.

The paper is organized as follows. First, we employ the basic relations in Section 2. In Section 3, the equivalent bending stiffness approach is carried out. Section 4 deals with homogenization of the basic relations. Section 5 presents an imperfection method. Finally, Section 6 presents concluding remarks.

## 2. A Statement of the Problem

The radius of the corrugated ring in the polar coordinate system is shown in Figure 1.

$$r = R(1 + hg(n\varphi)) \tag{1}$$

where $R$ is the radius of the basic circular ring, equally spaced from the crests of the corrugation, $h = \frac{H}{R}$, $H$ is the corrugation depth (Figure 2), $g(n\varphi)$ is the periodic function with period $2\pi/n$ (pitch of corrugation), $n$ is the number of corrugations, and $0 \leq \varphi \leq 2\pi, |hg| \leq 1$.

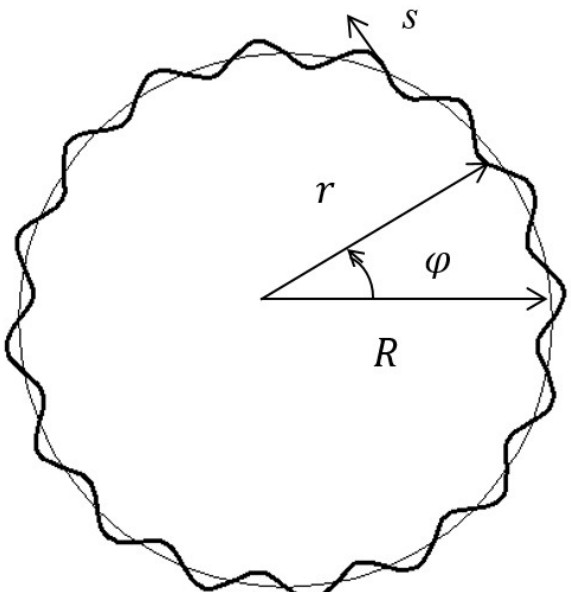

**Figure 1.** Corrugated ring with profile (4), $h = 0.05$; $n = 16$.

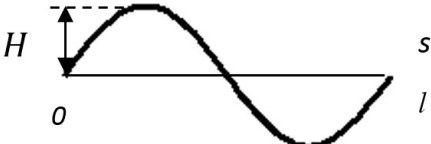

**Figure 2.** One pitch of corrugation.

In this case, arc $(\varphi_1, \varphi_2)$ length $s$ and radius of curvature $\rho$ of the ring are:

$$s = \int_{\varphi_1}^{\varphi_2} A d\varphi; \ \rho = \frac{A^3}{r^2 + 2r_\varphi^2 - rr_{\varphi\varphi}} \tag{2}$$

where $A = \sqrt{r^2 + r_\varphi^2}$, $(\cdot)_\varphi = \frac{d(\cdot)}{d\varphi}$.

We suppose

$$h \ll 1; \frac{2\pi}{n} \ll 1 \tag{3}$$

Further, we choose the corrugation profile most convenient for analysis (Figure 1):

$$r = R(1 + h cos(n\varphi)) \tag{4}$$

We assume that, for the ring, the hypothesis of flat cross-section is satisfied and the magnitude of the bending moment is proportional to the change in the curvature of the middle surface of the ring, in the section under consideration.

We deal with the general instability of the corrugated ring, i.e., the transition to the form with non-circular basic ring. Local buckling modes are absent. The following assumptions are also valid: spatial forms of equilibrium are not considered; during deformation the load remains directed normal to the deformed ring, and its intensity does not change.

The problem is solved in a linear formulation, using the static criterion of bifurcational stability loss. As shown in a number of studies [32–35], if the initial problem is conservative and the load is applied quasistatically, a similar approach to studying the stability of elastic systems is justified.

## 3. Equivalent Bending Stiffness Approach

Let us replace the corrugated ring with a circular one, radius R, with the equivalent bending stiffness. To determine this stiffness, following [16], we single out one corrugation wave, and due to the large number of such waves, the curvature of the basic circle can be neglected (Figure 2). Effective bending stiffness, D, is stiffness that ensures equality of deformations of the original curvilinear and approximating rectilinear beams. As a result, one obtains

$$D = EIk \tag{5}$$

where $k = \frac{1}{l_s} = \langle A \rangle^{-1}$, $\langle \ldots \rangle = \frac{1}{2\pi R} \int_0^{2\pi} (\ldots) d\varphi$; $E$ is the Young's modulus of the ring material; $I$ is the moment of inertia of the ring cross-section; $l_s = \int_0^{2\pi/n} A d\varphi$ and $l = 2\pi R/n$ are the lengths of the wave of the corrugation and its projection.

To evaluate the intensity of external pressure $p^0$ acting on an equivalent basic ring, we select arcs $ds$ and $dl$ (Figure 3). Projecting the load shows that the normal pressure on the basic ring is the same as the original one.

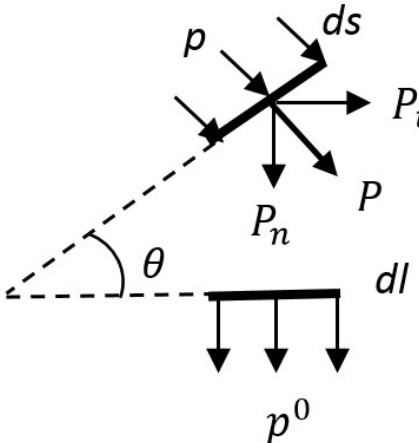

**Figure 3.** Projection of external pressure to the basic ring: $p$, $p^0$ are the intensities of external pressure acting on the corrugated and basic circular ring; $\theta$ is the angle between the tangents to the corrugated and the basic circlular ring; $ds$ and $dl$ are the arcs of the corrugated and basic circular ring, $dl = ds \cos\theta$; $P = pds = \frac{pdl}{\cos\theta}$; $P_n = P\cos\theta = pdl$; $p^0 = p$.

Note that the projection also gives a tangential load in the basic circular ring, self-balancing along one pitch of corrugation. Additional tangential load is not considered in the equivalent bending stiffness approach. Above, we deal with pressure which is permanently normal to the arc of the ring. If the external pressure is caused by radial forces directed permanently to the center (see [32], problem 134), then intensity $p$ must be replaced to $\frac{p}{k}$.

Buckling pressure for the circular ring with bending stiffness (5) is [34]

$$p_b = k \tag{6}$$

where $p_b = \frac{p_b^1}{p_b^0}$, $p_b^1 = \frac{3kEI}{R^3}$ is the buckling pressure of the circular ring with bending stiffness (5); $p_b^0 = \frac{3EI}{R^3}$ is the buckling pressure of the circular ring with radius, R, and bending stiffness *EI*.

Corrugation profile (4) has two important characteristics, namely, *n* and *h*. The buckling pressure (6) depends on the parameter *nh* (Table 1), which can be called the corrugation parameter. For $h = 0$, one has $k = 1$; by increasing *nh*, the critical pressure decreases (Figure 4).

**Table 1.** Buckling loads (6) for the circular ring with the equivalent bending stiffness for constant value of *nh*.

| *nh* = 1.6 | *n* = 160; *h* = 0.01 | *n* = 80; *h* = 0.02 | *n* = 40; *h* = 0.04 | *n* = 20; *h* = 0.08 | *n* = 16; *h* = 0.1 |
|---|---|---|---|---|---|
| $p_b$ | 0.6767 | 0.6767 | 0.6767 | 0.6766 | 0.6764 |

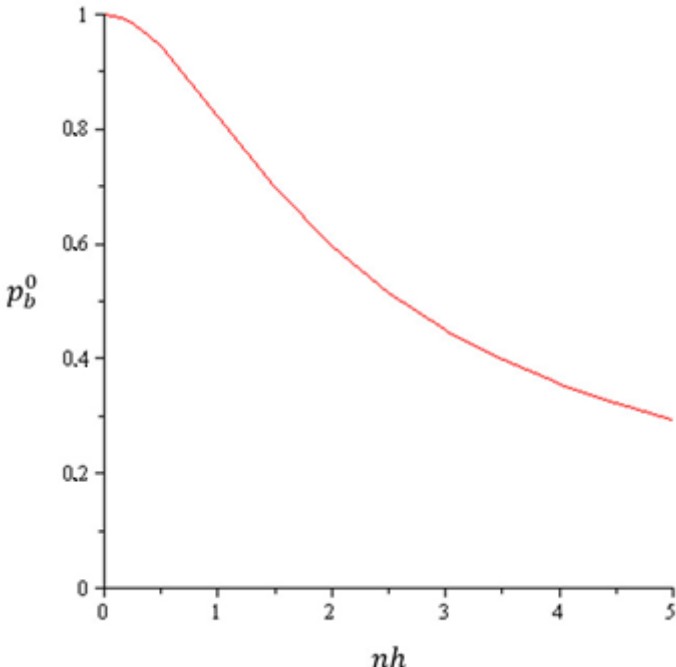

**Figure 4.** Ratio of buckling loads of corrugated (4) and basic circular rings $p_b^0$ on parameter $nh$ ($h = 0.05; n = 0\ldots100$).

Comparing of the corrugated and circular rings with the same perimeter shows that corrugation increases the critical pressure

$$p_{bp} = k^{-2} \tag{7}$$

where $p_{bp} = \frac{P_b}{P_{bp}}$, $P_{bp} = \frac{3EI}{(k^{-1}R)^3}$ is the buckling pressure of the circular ring of the same perimeter.

Radius $R_1$ of the circular ring, with the perimeter equal to the perimeter of the corrugated one, can be calculated as follows $R_1 = \int_0^{2\pi} A d\varphi / 2\pi$.

Using Formulas (6) and (7), one easily determines, for corrugation profile (4), that the buckling load of the corrugated ring is 31% higher than for the circular ring of the same perimeter. On the other hand, the corrugated ring restricts 23.6% less area. Comparison using (6) with the basic circular ring (Figure 1) shows a decrease in the critical pressure for the corrugated one by 12.6% and a decrease in the restricted area by 0.125%. In this case, the perimeter of the corrugated ring is 14.5% larger than the perimeter of the basic ring.

The questions remain unclear:

(a) the applicability of the equivalent bending stiffness approach to the stability problems;
(b) estimation of the error caused by neglecting the curvature of the basic circular ring in determining equivalent bending stiffness;
(c) estimation of the accuracy of the buckling pressure obtained with the equivalent bending stiffness approach and the possibility of its refinement.

## 4. Asymptotic Homogenization Method

Suppose that under the influence of an external pressure, the initial equilibrium state of the corrugated ring transits to a new equilibrium state, which has two perpendicular axes of symmetry. We write the equation describing this state and find the minimum value of pressure at which such a state becomes stable.

The symmetry axes of the new equilibrium form are taken as the x, y coordinate axes and consider the part of the ring lying above the horizontal axis of symmetry (Figure 5).

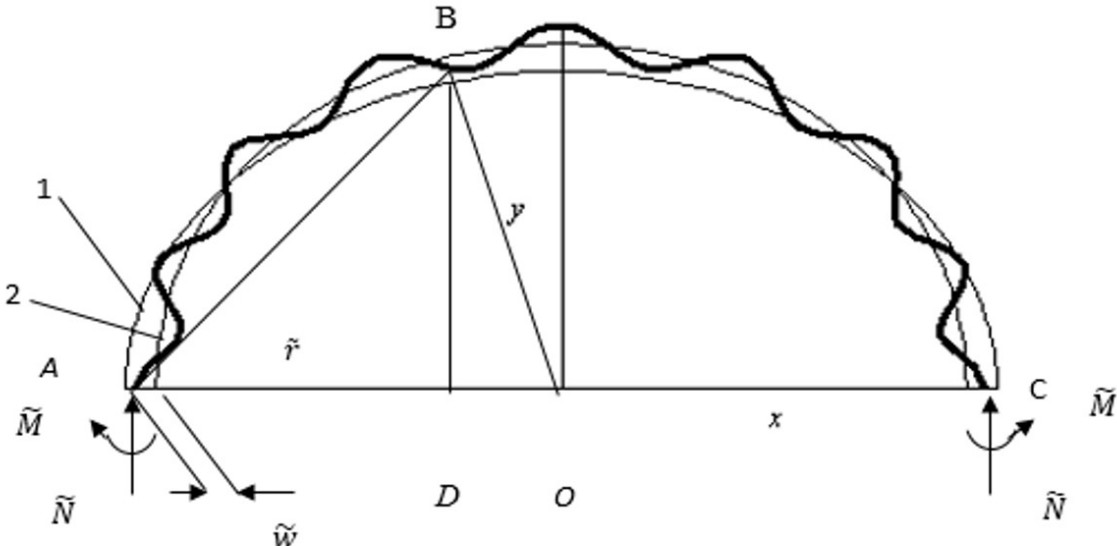

**Figure 5.** The part of the ring lying above the horizontal axis of symmetry: numbers 1 and 2 correspond to the middle lines of the basic circular ring before and after buckling; $\widetilde{N}$, $\widetilde{M}$, $\widetilde{w}$, $\widetilde{r}$ are the internal force, moment, deflection, and radius at the points A and C.

The bending moment $M$ in the cross-section $B$ is

$$M = \widetilde{M} + \widetilde{N}\left|\overline{AD}\right| - p\frac{\left|\overline{AB}\right|^2}{2} \tag{8}$$

where $\widetilde{N} = p(r - \widetilde{w})$.

Expression (8) can be written as

$$M = \widetilde{M} - p\left(\frac{\left|\overline{AB}\right|^2}{2} - \left|\overline{AO}\right|\left|\overline{AD}\right|\right) \tag{9}$$

From triangle ABO (Figure 5), one obtains

$$\frac{\left|\overline{AB}\right|^2}{2} - \left|\overline{AO}\right|\left|\overline{AD}\right| = \frac{1}{2}\left(\left|\overline{OB}\right|^2 - \left|\overline{OA}\right|^2\right) = \frac{1}{2}\left((r-w)^2 - (\widetilde{r}-\widetilde{w})^2\right) \tag{10}$$

Substituting expression (10) into Equation (9), after linearization with respect to w and $\widetilde{w}$, we obtain

$$M = \widetilde{M} - \frac{1}{2}p\left(r^2 - \widetilde{r}^2 + 2\widetilde{r}\widetilde{w} - 2rw\right) \tag{11}$$

We express the bending moment (11) through the projections of displacements on the basic circular ring (Figure 6)

$$M = EI\frac{1}{A}\left(\frac{1}{A}\left(\frac{ru + r_\varphi w}{\rho} - \left(\frac{rw - r_\varphi u}{A}\right)_\varphi\right)\right)_\varphi \tag{12}$$

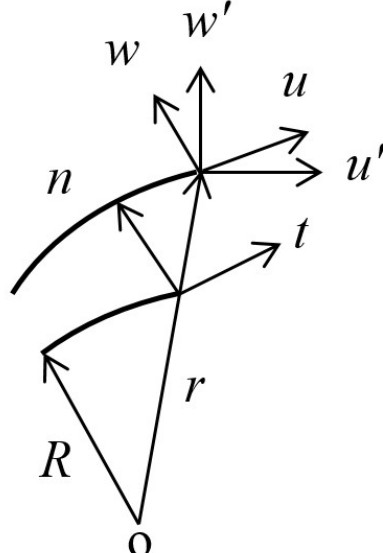

**Figure 6.** Projections of displacements on the basic circular ring. Here $u'$, $w'$ are the tangential and normal displacements of the corrugated ring; $u$, $w$ are the projections of these displacements on the basic circular ring; $u' = \frac{ru + r_\varphi w}{A}$; $w' = \frac{rw - r_\varphi u}{A}$.

Expression (12) is reduced to the form

$$M = EI\frac{1}{A}\left(\frac{1}{A^2}\left(r_\varphi \varepsilon - r\vartheta\right)\right)_\varphi \tag{13}$$

where

$$\varepsilon = u_\varphi + w; \ \vartheta = w_\varphi - u, \tag{14}$$

$\varepsilon$, $\vartheta$ are the circumferential deformation and the angle of rotation of the tangent to the basic circular ring [34].

We suppose inextensibility conditions for basic ring

$$u_\varphi + w = 0 \tag{15}$$

Taking into account condition (15), expression (13) can be rewritten as follows

$$M = EI\frac{1}{A}\left(\frac{r\left(w_\varphi - u\right)}{A^2}\right)_\varphi. \tag{16}$$

Substituting the expression for moment (16) into Equation (11), one obtains

$$\left(\frac{r\left(w_\varphi - u\right)}{A^2}\right)_\varphi + \bar{p}Arw = \frac{A}{2}\left(\bar{p}\left(r^2 - \widetilde{r}^2 + 2\widetilde{rw}\right) - 2\overline{M}\right) \tag{17}$$

where $\bar{p} = \frac{p}{EI}$; $\overline{M} = \frac{\widetilde{M}}{EI}$.

For eigenvalue problem (15), (17) must be supplemented with symmetry conditions (Figure 5)

$$w_\varphi(0) = w_\varphi\left(\frac{\pi}{2}\right) = 0 \tag{18}$$

So, we obtained an ODE with rapidly oscillating coefficients. This kind of problem can be efficiently solved by the homogenization method [36–38] based on the two-scale asymptotic expansions.

Let us introduce a new variable, $\xi = n\varphi$, which is assumed to be the independent on $\varphi$, hence

$$\frac{d}{d\varphi} = \frac{\partial}{\partial\varphi} + n\frac{\partial}{\partial\xi} \qquad (19)$$

The projections of the displacements $u$, $w$ are represented as series

$$u = \sum_{k=0}^{\infty} n^{-k}u_k(\varphi,\xi); \ w = \sum_{k=0}^{\infty} n^{-k}w_k(\varphi,\xi), \text{ corr} \qquad (20)$$

where $u_k$, $w_k$ are the $\xi$-periodic functions with period $2\pi$.

Substituting expansions (20) into Equations (15) and (17), after splitting in powers of $n^{-k}$, we obtain

$$\frac{\partial}{\partial\xi}\left(a(\xi)\frac{\partial w_0}{\partial\xi}\right) = 0 \qquad (21)$$

$$\frac{\partial u_0}{\partial\xi} = 0 \qquad (22)$$

$$\frac{\partial}{\partial\xi}\left(a(\xi)\frac{\partial w_1}{\partial\xi}\right) + \frac{\partial}{\partial\xi}\left(a(\xi)\left(\frac{\partial w_0}{\partial\varphi} - u_0\right)\right) = 0 \qquad (23)$$

$$\frac{\partial}{\partial\xi}\left(a(\xi)\frac{\partial w_2}{\partial\xi}\right) + \frac{\partial}{\partial\xi}\left(a(\xi)\left(\frac{\partial w_1}{\partial\varphi} - u_1\right)\right) + a(\xi)\frac{\partial^2 w_1}{\partial\varphi\partial\xi} + a(\xi)\frac{\partial}{\partial\varphi}\left(\frac{\partial w_0}{\partial\varphi} - u_0\right) + \bar{p}Arw_0$$
$$= \frac{A}{2}\left(\bar{p}\left(r^2 - \tilde{r}^2 + 2\widetilde{rw}\right) - 2\overline{M}\right) \qquad (24)$$

$$\frac{\partial u_0}{\partial\varphi} = -w_0 \qquad (25)$$

where $a(\xi) = \frac{r}{A^2}$.

Using Equations (21) and (22), one obtains

$$w_0 = w_0(\varphi); u_0 = u_0(\varphi) \qquad (26)$$

Then Equation (23) can be rewritten as follows

$$\frac{\partial}{\partial\xi}\left(a(\xi)\frac{\partial w_1}{\partial\xi}\right) = -\frac{\partial a(\xi)}{\partial\xi}\left(\frac{\partial w_0}{\partial\varphi} - u_0\right) \qquad (27)$$

Integrating Equation (27) over $\xi$, we obtain

$$\frac{\partial w_1}{\partial\xi} = -\frac{\partial w_0}{\partial\varphi} + u_0 + \frac{C(\varphi)}{a(\xi)} \ . \qquad (28)$$

The integration "constant" $C(\varphi)$ is determined from the periodicity condition for the function $w_1$ with respect to $\xi$:

$$C(\varphi) = \hat{a}\left(\frac{\partial w_0}{\partial\varphi} - u_0\right), \hat{a} = \left(\frac{1}{2\pi}\int_0^{2\pi} a^{-1}d\xi\right)^{-1} \qquad (29)$$

Excluding derivative $\frac{\partial w_1}{\partial\xi}$ from Equation (24), one obtains

$$\frac{\partial}{\partial\xi}\left(a(\xi)\frac{\partial w_2}{\partial\xi}\right) + \frac{\partial}{\partial\xi}\left(a(\xi)\left(\frac{\partial w_1}{\partial\varphi} - u_1\right)\right) + \hat{a}\frac{\partial}{\partial\varphi}\left(\frac{\partial w_0}{\partial\varphi} - u_0\right) + \bar{p}Arw_0$$
$$= \frac{A}{2}\left(\bar{p}\left(r^2 - \tilde{r}^2 + 2\widetilde{rw}\right) - 2\overline{M}\right) \qquad (30)$$

We apply to Equation (30) the averaging operator, $\int_0^{2\pi}(\cdots)d\xi$. The first two terms vanish due to the periodicity in $\xi$, and finally we have

$$\hat{a}\frac{d}{d\varphi}\left(\frac{dw_0}{d\varphi}-u_0\right)+\overline{p}\dot{s}w_0 = \frac{\overline{p}}{2}\left(\ddot{s}-\widetilde{sr}(\widetilde{r}-2\widetilde{w})\right)-s\overline{M} \tag{31}$$

where $\dot{s} = \frac{1}{2\pi}\int_0^{2\pi}Ard\xi$ ; $\ddot{s}=\frac{1}{2\pi}\int_0^{2\pi}Ar^2d\xi$ ; $s=\frac{1}{2\pi}\int_0^{2\pi}Ad\xi$ .

Using expression (25), Equation (31) is reduced to

$$\frac{d^2w_0}{d\varphi^2}+\beta^2w_0=\alpha \tag{32}$$

where $\beta^2=1+\frac{\overline{p}}{\hat{k}}$; $\hat{k}=\frac{\hat{a}}{\dot{s}}$; $\alpha=\frac{\overline{p}}{2\hat{a}}\left(\ddot{s}-\widetilde{sr}\left(\widetilde{r}-2\widetilde{w}\right)\right)-\frac{s\overline{M}}{\hat{a}}$.

The general solution of the ODE (32) is

$$w_0 = C_1\sin\beta\varphi+C_2\cos\beta\varphi+\frac{\alpha}{\beta^2} \tag{33}$$

where $C_1$, $C_2$ are the integration constants.

For function $w_0$, one has

$$\frac{dw_0}{d\varphi}=0 \text{ at } \varphi=0,\ \frac{\pi}{2} \tag{34}$$

Substituting expression (33) into Equation (34), we find

$$C_1=0; \beta=2m, m=1,2,\ldots \tag{35}$$

Assuming $m=1$, one finds from expression (35) the buckling pressure of the corrugated ring

$$\overline{p}_b^2 = 3\hat{k} \tag{36}$$

Thus, the rationale for the use of the equivalent bending stiffness approach to the problem of the stability of a corrugated ring is obtained.

Figure 7 shows how the value of the critical pressure of the corrugated ring $\overline{p}_b^2$ decreases compared to a value for the circular ring of radius $R$ with an increase in the depth of the corrugation $h$.

Thus, an increase in the depth of the corrugation decreases the critical pressure for the corrugated ring.

Figure 8 shows a comparison of buckling pressures $\overline{p}_b^2$ and $\overline{p}_b^{2e}$ for corrugated and circular rings having the same volume of material. It is obvious that it is more optimal to increase the buckling pressure by increasing the cross section of a circular ring than by corrugating it.

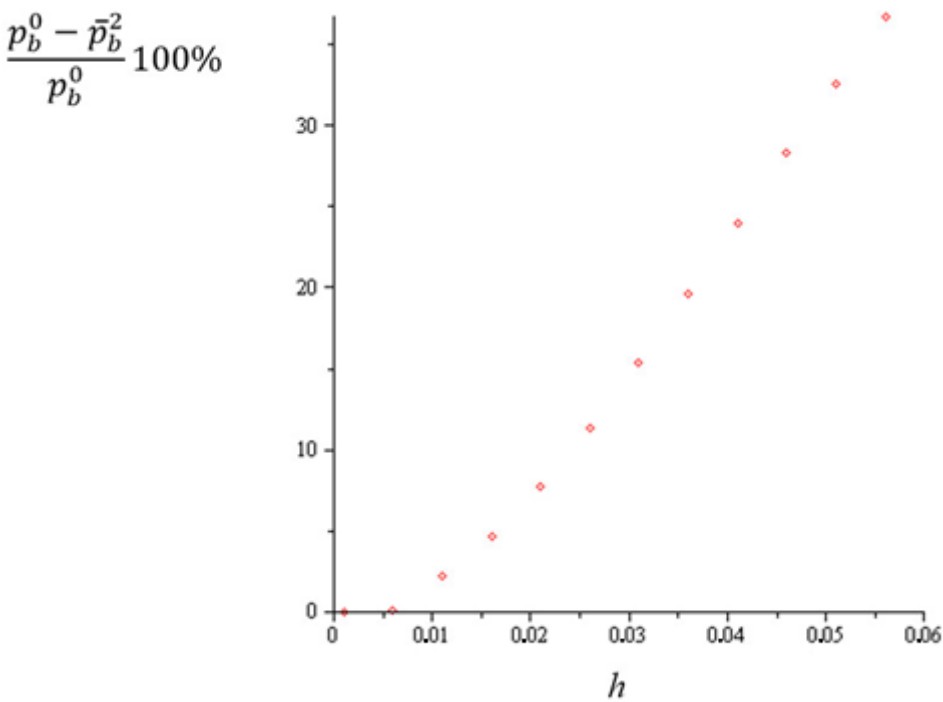

**Figure 7.** Decrease of the buckling pressure of a corrugated ring with an increase in the depth of the corrugation.

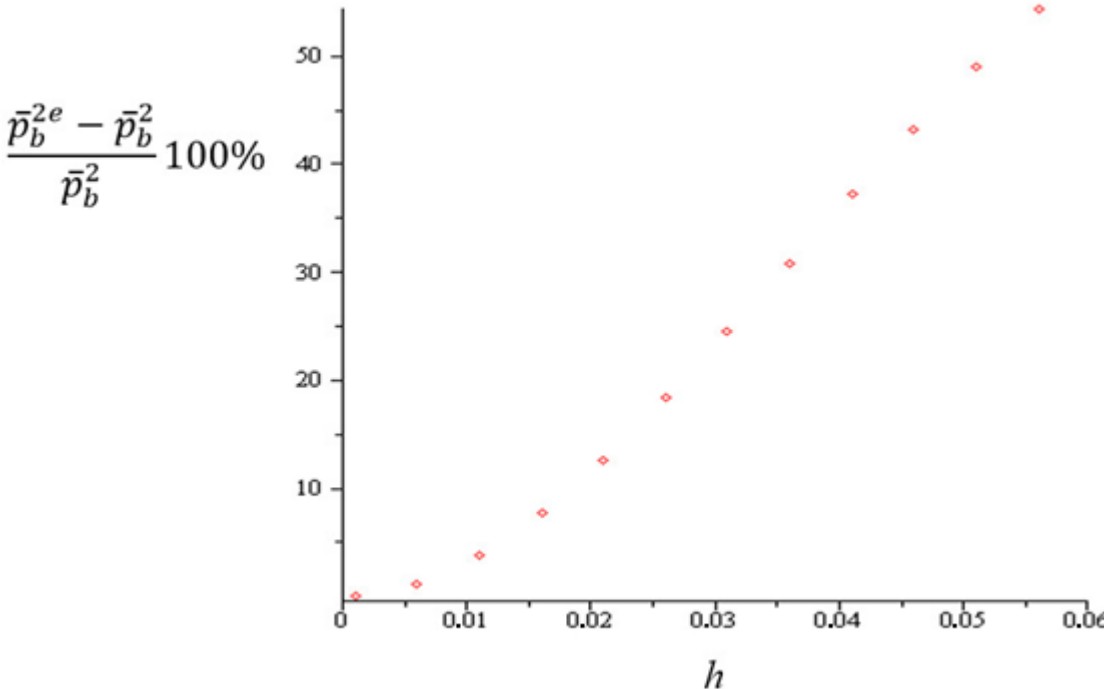

**Figure 8.** Comparison of buckling pressures $\bar{p}_b^2$ and $\bar{p}_b^{2e}$ for corrugated and circular rings having the same volume of material.

Asymptotic homogenization method gives less of a value of a buckling load than equivalent bending stiffness approach. This difference increases with the increase of the depth of the corrugation $h$ (Figure 9).

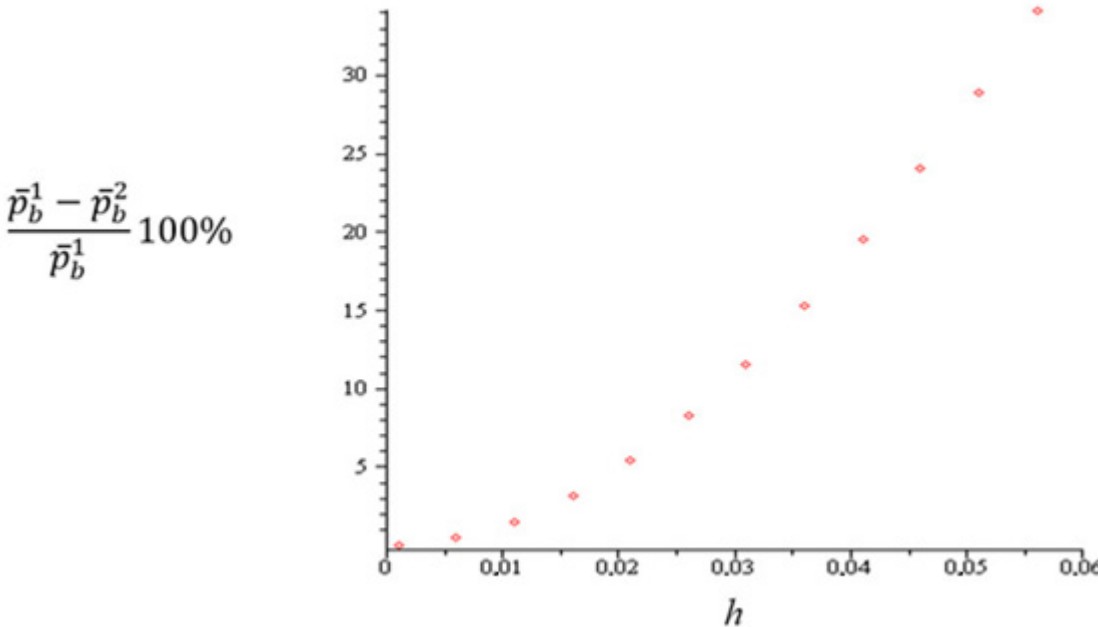

**Figure 9.** Comparison of buckling pressure obtained using the asymptotic homogenization method (36) and equivalent bending stiffness approach (7).

The question of estimating the accuracy of the results obtained using the equivalent bending stiffness approach and asymptotic homogenization method remains open.

## 5. Using the Imperfection Method

"The imperfection method is characterized by the question: What is the value of the load for which the static displacements of the imperfect system become excessive or even infinite?" [33]. This approach was widely used by Timoshenko [34]. Critical re-examination of the imperfection method was described by Gubanova and Panovko [35].

Let us write the ODE of the deformed midline of the ring in the framework of the flat cross-section hypothesis

$$\frac{1}{\rho} - \frac{1}{\overline{\rho}} = \frac{M}{EI} \tag{37}$$

where $\frac{1}{\rho} = \frac{(r-w)^2 + 2(r-w)_\varphi^2 - (r-w)(r-w)_{\varphi\varphi}}{\overline{A}^3}$; $\overline{A} = \sqrt{(r-w)^2 + (r-w)_\varphi^2}$.

Substituting the expansion $\overline{A}$ in powers of $\frac{w}{R}$ and expression (11) for the bending moment $M$ to the equilibrium Equation (37), after ignoring the second-order terms, one obtains

$$\begin{aligned}
rw_{\varphi\varphi} + \tfrac{3}{A}\left(rr_\varphi + 2\tfrac{r_\varphi^3}{r} - \tfrac{4}{3}r_\varphi A - r_\varphi r_{\varphi\varphi}\right)w_\varphi \\
+ \tfrac{1}{A}\left(Ar_{\varphi\varphi} - 2rA + 3r^2 + 6r_\varphi^2 - 3rr_{\varphi\varphi} + prA^4\right)w \\
= \left(\overline{p}\left(r^2 - \widetilde{r}^2 + 2\widetilde{r}\,\widetilde{w}\right) - 2\overline{M}\right)\tfrac{A^3}{2}
\end{aligned} \tag{38}$$

The solutions of Equation (38) must satisfy the symmetry conditions (18).

Numerical solutions of the boundary value problem (38), (18) were obtained on the basis of the following iterative scheme: at the first iteration, we suppose $\widetilde{w} = 0$; $\overline{M} = 0$ and calculate $\widetilde{w} = w(0)$; $\overline{M} = \frac{1}{\rho(0)} - \frac{1}{\overline{\rho}(0)}$. Then, new values of $\widetilde{w}$ and $\overline{M}$ are substituted into Equation (39) and used in the second iteration. Subsequent iterations are carried out similarly. This algorithm was used for the following corrugation profile

$$r = 1 + h\cos(2\pi n\varphi) \tag{39}$$

Symmetry conditions are

$$w_\varphi(0) = w_\varphi\left(\frac{1}{4}\right) = 0 \tag{40}$$

Comparison of the results of calculation of the deflection at the point $\varphi = 0$ for $n = 16$, $h = 0.01$, showed a satisfactory convergence of the used iterative scheme (Table 2).

**Table 2.** Comparison of the results of calculation of the deflection at the point $\varphi = 0$ for $n = 16$, $h = 0.01$; here $\Delta_i = \frac{w_{i-1} - w_1}{w_{i-1}} 100$, $i = 1 - 4$.

| Number of Iteration/Pressure | $p = -1$ | $p = -1.5$ | $p = -2$ |
|---|---|---|---|
| 3 | $\Delta_3 = 0.8\%$ | $\Delta_3 = 1.09\%$ | $\Delta_3 = 12\%$ |
| 4 | $\Delta_4 = 0.01\%$ | $\Delta_4 = 1.06\%$ | $\Delta_4 = 3.18\%$ |

To estimate the critical pressure of the corrugated ring, the dependences of the deflection at the point were $\varphi = 0$ on the external pressure for corrugation profile (39) with parameters $n = 16$, $h = 0.01$ (Figure 10a); $n = 8$, $h = 0.02$ (Figure 10b).

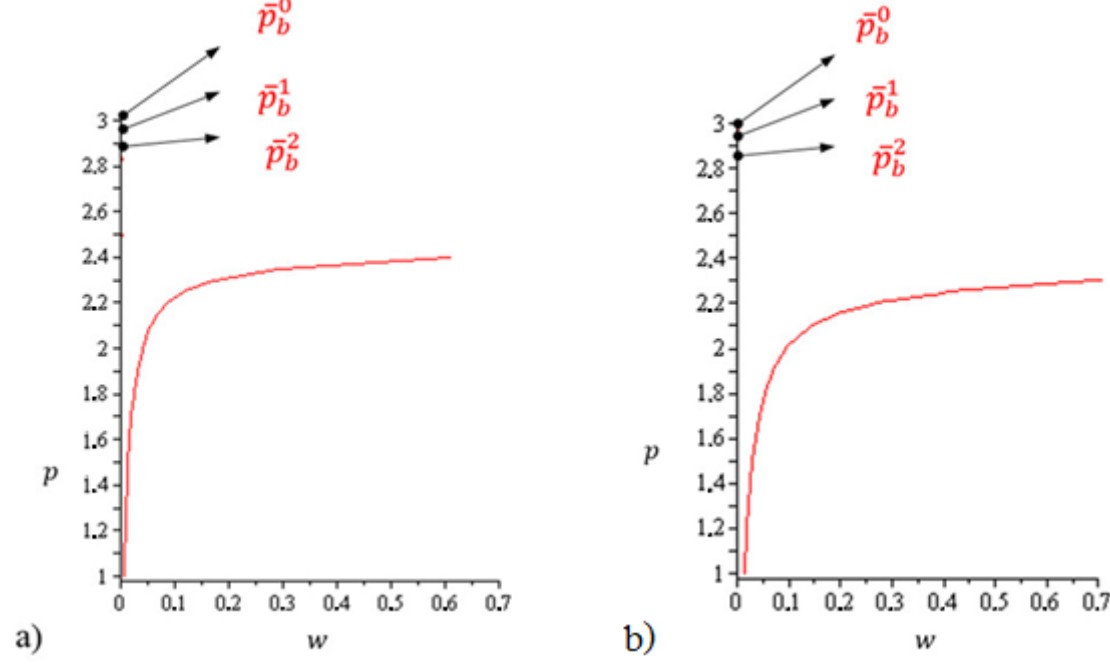

**Figure 10.** Diagrams of loading a corrugated ring with a corrugation (39): (**a**) $n = 16$, $h = 0.01$; (**b**) $n = 8$, $h = 0.02$; $\bar{p}_b^0 = 3$, $\bar{p}_b^1 = 2.98$, $\bar{p}_b^2 = 2.94$ are buckling pressures for the smooth ring and for the corrugated ring obtained by the equivalent bending stiffness approach and asymptotic homogenization method.

The pressure at which a sharp increase in the deflection occurs gives estimation of buckling load. The same figures show that the buckling pressure values found using the equivalent bending stiffness approach $\bar{p}_b^1$, asymptotic homogenization method $\bar{p}_b^2$, and buckling pressure for circular ring $\bar{p}_b^0 = 3$.

## 6. Conclusions

One can increase the critical pressure of a circular ring of a given perimeter using corrugation. However, a corrugated ring restricts less area, which is of practical importance in the design of cylindrical tanks. Comparison with the basic circular ring shows a decrease in critical pressure. Thus, long cylindrical circumferentially-corrugated shells have less rigidity at external pressure than circular shells with basic radius.

It is well known [39–41] that initial imperfections lead to the decrease of critical pressure for shells. However, in this case, one deals with initial imperfections whose change in circumferential direction coincides with a buckling form of the ideal shell. A circular ring buckles along two waves; therefore, many-waves corrugation increases its stability.

We solved our problem under the assumption that the pre-buckling state is momentless. For circular cylindrical shells under axisymmetric transverse pressure, this assumption is correct [42]. In our case, the correctors to the homogenized solution are rapidly variable, therefore, their projection onto the slow changing buckling form is insignificant.

The critical pressure of the corrugated ring depends on the product of two parameters of the corrugation: number of corrugations *n* and their depth *h*. Rings with the same values of parameter nh, have the same critical pressure.

The asymptotic homogenization method gave the possibility of mathematically justifying the equivalent bending stiffness approach and corrected the values of reduced bending rigidity and buckling pressure.

In our paper, corrugation profile has a cosine-shaped configuration. In engineering practice, a lot of other configuration, such as trapezoidal, rectangular, and zigzag, profiles are used. The main conclusions that we obtained based on the analysis of the simplest case remain unchanged. Namely: the corrugation reduces the critical pressure, the decrease depends mainly on the corrugation parameter nh, with increasing nh, and the critical pressure decreases.

Asymptotic homogenization method is applicable for the above profiles according to the scheme developed in this paper. Only the solutions to the cell problem will change, and therefore the values of equivalent stiffnesses will be different.

Replacing the original corrugated ring with a circular one of the same perimeter leads to a symmetrization of the problem. It was shown in [43] that an increase in symmetry in eigenvalue problems for ODEs with periodic coefficients leads to an increase in eigenvalues. Therefore, the found-above values of buckling loads are overestimated.

A few words concerning possible generalizations: experimental studies [7] indicate the need to take into account, in many cases, inelastic general instability and local buckling modes. When deriving nonlinear equations by taking into account inelastic deformation, the results obtained in [44,45] can be used. The models proposed in [25,26] could be used for improving the original model, accounting more exactly for the coupling between tangential and bending loads.

**Author Contributions:** Software, I.I.A. and E.V.R.; methodology, I.V.A.; formal analysis, A.A.D. All authors discussed all stages of the research and preparing of the manuscript. All authors have read and agreed to the published version of the manuscript.

**Funding:** This research received no external funding.

**Acknowledgments:** The research and publication of this article were funded by the authors themselves. The authors thank the anonymous referees whose valuable comments and suggestions favored improvement of the paper.

**Conflicts of Interest:** The authors declare that there is no conflict of interest regarding the publication of this paper.

**Data Availability:** All the data required for the publication have been included in the research article. These are in the form of tables and plots in the research paper.

## Abbreviations

The following abbreviations are used in this manuscript:

ODE     Ordinary Differential Equation
FEA     finite element analysis

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
