# Peer review of "Buckling of Corrugated Ring under Uniform External Pressure"

_symmetry, doi:10.3390/sym12081250_

Round 1

Reviewer 1 Report

The Corrugated ring structure under buckling external pressure was proposed in this paper. The three bucking theories were used for the stability analysis. This can be interesting work, outlining the first principle of the static buckling analysis of different types of ring structures. However, some issues are raised in the paper that need further explanations.

Specific remarks:    

  1. Critical load is one of the essential studies of the buckling problem. It shows how the structure changes under such condition. For elasticity problem, the geometrical nonlinearity can occur on the ring structure when the critical pressure is reached. But, there is no proof how to calculate and predict the onset of buckling analysis for this paper.
  2. On page 5, it is said that the buckling load of the corrugated ring is 31% higher than for circular ring of the same perimeter; but corrugated ring restricts 23.6% less area. Comparison with the basic circular ring shows a decrease in the critical pressure for the corrugated one by 12.6% and a decrease in the restricted area by 0.125%. Where does this statement come from?. Is it assumption or based on the calculations ?.
  3. The corrugation profile is a cosine-shaped configuration. Why choose this?. Are there any benefits of this profile?. What about other configurations such as trapezoidal, rectangular, and zigzag profiles ?
  4. Amongst the three theories, which one is more effective use for the stability analysis on the buckling problem?. 
  5. The stability analysis due to the buckling dynamic analysis under increasing frequency can be valuable to predict the limit of pressure load onto ring structure. Why this is not given on the paper.
  6. Insert Reference for the equation in the Introduction section.
  7. 5 with different colour of the ring profile needs to be labelled properly. Also, need to clearly explain on the paper. Caption in Fig. 13, where is (54) on the text?
  8. The review in this paper is short time. Check carefully the notations used for some equations. also, define carefully any variables or parameters.
  9. English editing can be helpful throughout this paper. Each passage and paragraph of the paper must be carefully rewritten. Only a few of problems as examples are shown below (remaining parts must be checked). 

Page 11: “Thus, long cylindrical corrugated in the circumferential direction shells have…” (should be rewritten as, “Thus, long cylindrical corrugated shells in the circumferential direction have..”).

Page 5, Line 1: “…then…” Page  2 line 1:… sence…” Page 1:”… anisotropic..” Page 1: “… cilyndrical…”Page 1: “…. a simpler problem is solved - the calculation ….pressure” (need to rewrite this sentence). Page 10:”… estimazion…”

Author Response

Our answers in attached file.

Reviewer 2 Report

The paper under review deals with stability analysis of corrugated ring subjected to external pressure. The problem is solved in three ways: equivalent bending stiffness approach; asymptotic homogenization method; the imperfection method. The use of the asymptotic homogenization method allows to mathematically justify the results. Then, The influence of the corrugation parameters on the buckling pressure is analyzed.

The paper is a good piece of work and present some originality aspects. I would suggest the author to comment and stress the influence of the corrugation function g in the results. 

According to the reviewer opinion, the paper can be published after a minor revision.

Author Response

Our responce in attached file.

Reviewer 3 Report

This paper is really well-written and I liked the presentation

The deduction of the basic equations is elegant and willingly minimalistic 

However it has to be remarked that they can be deduced from a more general setting 

(see 

Dell'Isola, Francesco, and A. Romano. "On the derivation of thermomechanical balance equations for continuous systems with a nonmaterial interface." International Journal of Engineering Science 25.11-12 (1987): 1459-1468.)

in a very similar way as done in

(Dell'Isola, Francesco. "Linear Growth of a Liquid Droplet Divided from its Vapour by a" SOAP BUBBLE"-like Fluid Interface." (1989).)

Minor improvements of the English style are required 

Author Response

Our answers in attached file.

Author Response

Our answers are in attached file.

Round 2

Reviewer 1 Report

The author has revised the manuscript, and the response to my previous comments was also given. Again, I would suggest the author checks carefully typos in the paper.

  • Radius R_1: should be R1 ?
  • some sense stiffness: why add “sense”. Doesn’t give any meaning.
  • The question of estimation the accuracy: should be “The question of estimating the …..”
  • widly :you mean “widely” ?
  • ……in the article Only the solutions…..: Add punctuation mark, full stop (….article. Only….)
  • …..nonlinear equations taking into account…..: should be “….equations by taking….
  • etc

Reviewer 4 Report

The paper still needs some improvement, according to the list reported in the following.
1. Page 1 At the end of the page, a locution in Cyrillic green font is reported, but it seems meaningless.
2. Page 2 The sentence “In paper [7] describes the nonlinear . . . ” does not have a subject.
3. Page 2 The paragraph “In paper [7] describes the nonlinear . . . ” is just one sentence - less than two rows- long. Reconsider the paragraph subdivision.
4. References [25] and [26] are not cited in the document. In reply to reviewer, authors stated that the mentioned modeling approaches could be used in future work, such a sentence has to be added at the end of Section 6.
5. Page 6 Typo “R 1” use subscript.
6. When referring to a numbered object Equation has to use capital letter “Equation (9)” and not “equation (9)”. Check the whole document.
7. Page 14 Typo: “and” in plain font and not slanted.
